# Renal Side Effects of COVID-19 Vaccination

**DOI:** 10.3390/vaccines10111783

**Published:** 2022-10-23

**Authors:** Junfeng Zhang, Jiajia Cao, Qing Ye

**Affiliations:** Department of Clinical Laboratory, The Children’s Hospital, Zhejiang University School of Medicine, National Clinical Research Center for Child Health, National Children’s Regional Medical Center, Hangzhou 310052, China

**Keywords:** COVID-19 vaccination, side effect, adverse reaction, kidney disease, renal, SARS-CoV-2

## Abstract

**Background**: The COVID-19 pandemic has imposed a challenge on global healthcare and has tremendously impacted everyone’s lives. Vaccination is one of the most effective and vital strategies to halt the pandemic. However, new-onset and relapsed kidney diseases have been reported after COVID-19 vaccination. This narrative review was conducted to collect published data and generalize some hypotheses for the pathogenesis of renal side effects of COVID-19 vaccines. **Methods**: A systematic literature search of articles reporting renal adverse reactions, including in adults and children, in the PubMed and Web of Science databases until August 2022 was performed. **Results**: A total of 130 cases reporting a renal adverse reaction following COVID-19 vaccination from 90 articles were included in this review, of which 90 (69%) were new-onset kidney diseases, while 40 (31%) were relapsed kidney diseases. The most frequent renal side effects of COVID-19 vaccination were minimal change disease (52 cases), IgA nephropathy (48 cases), antineutrophil cytoplasmic autoantibody vasculitis (16 cases), and acute interstitial nephritis (12 cases). Other renal side effects occurred at a much lower frequency. Follow-up data were available for 105 patients, and 100 patients (95%) responded to the treatments. **Conclusions**: The number of reported cases is far less than the hundreds of millions of vaccinations, and the benefit of COVID-19 vaccination far outweighs its risks. This review will assist healthcare professionals, particularly nephrologists, who should be aware of these side effects and recognize them early and treat them efficiently.

## 1. Introduction

With the ongoing COVID-19 pandemic, vaccination programs are being rolled out worldwide to prevent COVID-19 and alleviate the severity of the disease. Over the past two years, multiple COVID-19 vaccines have been granted emergency use authorization, including mRNA vaccines (such as Pfizer-BioNTec and Moderna), adenovirus-vectored vaccines (such as Oxford-AstraZeneca and Janssen), recombinant vaccines (Novavax), and inactivated vaccines (CoronaVac) [1].

At present, short-term side effects are usually mild and self-limiting and mostly involve local injection symptoms such as pain, swelling and urticarial eruptions, and systemic symptoms of fever and headache [1,2]. However, since mass scale vaccination, a growing number of severe and required hospitalization adverse reactions to vaccinations has been reported, including neurological side effects [3], myocarditis [4], and autoimmune disease [5]. Furthermore, newly diagnosed, or relapsed kidney diseases have been reported in adults and children [6,7,8].

Given the emerging evidence between kidney diseases and COVID-19 vaccination, the risk of renal side effects has sparked public concern. Here, this narrative review was conducted to collect the renal side effects in the published data and discuss the plausible mechanism of action triggered by COVID-19 vaccination.

## 2. Methods

A systematic literature search was performed in the PubMed and Web of Science databases before August 2022 using the search terms “COVID-19 Vaccine” OR “COVID-19 Vaccination” OR “ SARS-CoV-2 vaccine” OR” SARS-CoV-2 Vaccination” AND “kidney” OR “Renal”. After the search was complete, all duplicates were removed and papers reporting the same case were also excluded. The titles and abstracts of the remaining studies were reviewed to ensure that they presented the association between renal disease and COVID-19 vaccination. Then the full-text screening, based on the relevance of the renal side effects of COVID-19 vaccination was selected. We only selected case or case series reports, other types of articles were excluded. Case or case series reporting new onset or relapse kidney histopathology in both native and transplanted kidneys following COVID-19 vaccination were included, and patients who had suffered from COVID-19 were excluded. Finally, 90 papers and 130 cases that developed renal side effects (128 cases of native kidneys and 2 cases of transplanted kidneys) were included in this systematic review [9,10,11,12,13,14,15,16,17,18,19,20,21,22,23,24,25,26,27,28,29,30,31,32,33,34,35,36,37,38,39,40,41,42,43,44,45,46,47,48,49,50,51,52,53,54,55,56,57,58,59,60,61,62,63,64,65,66,67,68,69,70,71,72,73,74,75,76,77,78,79,80,81,82,83,84,85,86,87,88,89,90,91,92,93,94,95,96,97,98]. We then extracted patient demographics (age and sex), medical history, vaccine type, number of vaccine doses given, baseline characteristics, laboratories upon presentation, onset of symptoms, timing of symptom onset, treatments, and outcomes. The study selection process was carried out using the Preferred Reporting Items for Systematic Reviews (PRISMA) guidelines (Figure 1).

Continuous data are presented as medians and ranges, while categorical data are presented as numbers and percentages.

## 3. Results

### 3.1. Overall

The renal side effects of COVID-19 vaccination collected from the literature mainly include minimal change disease (MCD), IgA nephropathy (IgAN), antineutrophil cytoplasmic autoantibody (ANCA) vasculitis, and acute interstitial nephritis (AIN). A total of 128 patients are summarized in Table 1, including 89 (70%) cases of newly diagnosed renal involvement and 39 (30%) cases of relapse. Of these, 52 (41%) cases were diagnosed with MCD, which is by far the most frequent renal side effect of COVID-19 vaccination, and the second most common pathology was IgAN (48/128, 37.5%), followed by 16 (12.5%) ANCA and 12 (9.0%) AIN. The median age was 42 (range 12–85) years, and 53% (68 of 128 cases) of patients were male.

Some others included membranous nephropathy (MN) in 8 cases, anti-glomerular basement membrane (anti-GBM) nephritis in 3 cases, focal segmental glomerulosclerosis (FSGS) in 3 cases, lupus nephritis in 3 cases and granulomatous vasculitis in 1 case. (Table 2) Renal side effects develop after any of the commercially available COVID-19 vaccinations, but 56% (72/128) of patients received the BNT162b2 (Pfizer) vaccine, followed by 30% (38/128) receiving the mRNA-1273 (Moderna) vaccine. In addition, 10.2% (13/128) received adenovirus vector (AstraZeneca) vaccine, 1.5% (2/128) received adenovirus vector (Janssen) vaccine, and another 2.3% (3/128) received inactivated vaccine (CoronaVac). Of these, 39% (50/128) of patients developed symptoms after the first dose, while 61% (78/128) of patients developed symptoms after the second dose. However, most reports did not describe whether the second dose of COVID-19 vaccines was the same as the first dose.

### 3.2. Minimal Change Disease (MCD)

MCD was the most common pathological type of renal side effect reported following COVID-19 vaccination, with a total of 52 cases (38 new and 14 relapse). The median age was 44 (14–83) years, with a male predominance (31/52, 60%). Approximately 28.8% of patients did not have any medical history of chronic illness, while 21% of patients had a history of hypertension, diabetes, or dyslipidemia. However, 76.9% (40/52) of patients developed edema, which is the most common symptom after vaccination. A total of 28 (53.8%) patients had clinical symptoms after the first dose of vaccine, of which 22 were new cases and 6 were relapses, with a median onset time of 7 (1–46) days. On the other hand, 24 (46.2%) patients presented after the second dose of vaccine, of which 16 were new cases and 8 were relapses, with a median onset time of 8 (2–88) days. Thirty-nine (75%) patients received steroid therapy, 5 patients received steroids and cyclophosphamide or mycophenolate mofetil or cyclosporine combination immunosuppression therapy, 3 patients underwent plasmapheresis or hemodialysis, 3 patients received rituximab, and the other 2 patients received conservative treatment. Follow-up data were available for 42 patients; in total, 41 (97.6%) patients responded well (28 patients achieved complete remission, 6 patients achieved partial remission, 7 patients improved) to the treatments, and 1 had no response [11] (Table 1).

### 3.3. IgA Nephropathy (IgAN)

IgAN was the second most frequent complication of renal side effects after COVID-19 vaccination. There were 48 patients with IgAN reported in the literature, of which 29 (60.4%) were new cases and 19 (39.6%) were relapse, with a median age of 33 (12–79) years, and approximately 64.6% of cases had a history of abnormal urine or kidney disease. The most frequent manifestation following COVID-19 vaccination was gross hematuria (43/48, 89.6%). In addition, 14 (29.2%) patients presented with fever, headache, nausea, vomiting, anorexia, or diarrhea, and 10 (20.8%) patients presented with acute kidney injury or renal failure, while proteinuria was observed in 7 (14.6%) cases. Most cases (35/48, 72.9%) occurred after the second dose, of which 19 were new cases and 16 were relapses, and the median time between COVID-19 vaccination injection and onset of symptoms was 2 (1–42) days. In contrast, 13 (27.1%) patients who developed clinical symptoms were reported following the first dose, of which 10 were new cases and 3 were relapses, with a median onset time of 4 (1–61) days. In these cases, 62.5% of patients received the Pfizer vaccine, 31.3% of patients received the Moderna vaccine, 4.2% of patients received the AstraZeneca vaccine, and the other 2.1% received CoronaVac. Twelve out of 48 patients (25.0%) were managed with steroids, 4 received combination immunosuppression therapy, and 3 patients received plasmapheresis or hemodialysis. Approximately half of the patients 24 (50.0%) were treated conservatively, and the other 4 (8.3%) patients symptoms subsided spontaneously. Forty patients had available follow-up data; of these, 37 (92.5%) patients responded to the treatments, and 3 had no response [41,47,93] (Table 1).

### 3.4. Anti-Glomerular Basement Membrane (Anti-GBM) and Anti-Neutrophil Cytoplasmic Autoantibody (ANCA) Vasculitis

There were ten cases with new-onset ANCA vasculitis, including 3 cases associated with proteinase 3 (PR3) and 7 cases of myeloperoxidase (MPO)-associated vasculitis, while six cases were ANCA vasculitis relapses. Eleven patients with ANCA vasculitis developed symptoms after the second dose of vaccine, with a median onset time of 14 (1–60) days, and 5 patients developed symptoms after the first dose, with a median onset time of 7 days. The clinical presentation was acute kidney injury or renal failure in 3 cases, 2 cases presenting with hematuria and 2 cases with proteinuria. Based on kidney biopsy, all ten new-onset patients confirmed crescentic glomerulonephritis and were treated with standard immunosuppression, of which 5 patients received steroids combined with plasmapheresis or hemodialysis, and 5 patients received combination immunosuppression therapy (4 patients were treated with rituximab and steroids). Eleven patients had available follow-up data, and 10 improved (Table 1).

Three cases of new-onset anti-GBM nephritis have been reported following COVID-19 vaccination. One case had no medical history and presented with fever, anorexia, and gross hematuria two weeks after the second dose of the Moderna vaccination, and the other case with a past medical history of hyperlipidemia developed macroscopic hematuria a day after the second dose of the Pfizer vaccination. In both cases, kidney histopathology revealed diffusely crescentic glomerulonephritis as well as linear staining of GBM for IgG. The two patients developed acute kidney injury and received immunosuppression therapy and plasma exchange; one patient remained on dialysis [54], while the other patient had no follow-up records [70]. The third patient was newly diagnosed with atypical anti-GBM nephritis accompanied by hypertension one week after the first Pfizer vaccination. The patient had not yet responded to immunosuppression therapy, and his serum creatinine level continued to rise [41] (Table 2).

### 3.5. Acute Interstitial Nephritis (AIN)

In the literature, 12 patients with a median age of 44 (12–77) years developed new-onset AIN, of which 8 patients developed symptoms following the second administration of vaccination with a median onset time of 14 (2–42) days, 4 patients developed symptoms after the first dose, and the median onset time was 14.5 (2–28) days. All 12 patients presented with acute kidney injury, and 5 patients presented with fever, anorexia, nausea, vomiting, or pain. Seven (58.3%) patients received steroid therapy, 4 (33.3%) patients were treated with hemodialysis, and they responded to the treatments. In addition, one patient with renal insufficiency gradually improved with supportive care (Table 1).

### 3.6. Membranous Nephropathy (MN)

MN developed in 8 patients, including 3 relapse patients who were associated with serum anti-phospholipase A2 receptor (PLA2R) antibody positivity and 5 new-onset patients, of which 1 new patient had neural epidermal growth factor-like 1 protein (NELL-1)-associated MN. Three new-onset patients developed nephrotic syndrome, of which 1 patient was treated with glucocorticoids, 2 patients received rituximab, and outcome data were not reported in these 3 patients [67]. One new-onset patient developed edema managed with conservative treatment, and no signs of spontaneous remission were observed within 60 days [83]. The other case of new-onset NELL-1-associated MN was treated conservatively with angiotensin converting enzyme inhibitor, and proteinuria significantly improved from 6.5 g/d to 0.4 g/d. [41]. Of 3 patients with PLA2R-associated MN, only 1 patient responded to tacrolimus with proteinuria, and serum albumin improved; 1 patient received obinutuzumab, but the outcome was unknown [41], and the other patient had no records of treatment or follow-up [55] (Table 2).

In addition, Gueguen et al. [84] reported a new case that developed PLA2R-associated MN after the first dose of Pfizer vaccination and achieved partial control after treatment with renin-angiotensin system blockade. However, edema worsened after the second dose of Moderna vaccination, and the patient achieved partial remission with rituximab treatment.

### 3.7. Others

There were 2 new-onset cases of focal segmental glomerulosclerosis (FSGS). One patient was in partial remission with persisting proteinuria and hyperlipoproteinemia after prednisolone therapy [37], and the other patient showed a tip-variant FSGS lesion that received glucocorticoid treatment with no follow-up data [67]. One patient with previous MCD but on repeat biopsy revealed a tip-variant FSGS lesion and achieved partial response to prednisone combined with tacrolimus therapy [41].

Kim et al. [91] reported a case of new-onset class III lupus nephritis with multiorgan involvement, and Zavala-Miranda et al. [82] reported a new-onset systemic lupus erythematosus beginning as class V lupus nephritis in a 23-year-old woman. The clinical symptoms of the 2 patients improved after immunosuppression treatmen**t**.

Tuschen et al. [81] reported a known case of class V lupus nephritis in remission presented with a flare up of lupus nephritis a week following the first dose of Pfizer vaccination. The patient received immunosuppressive therapy with mycophenolate mofetil and prednisolone, but her proteinuria was slow to resolve.

Gillion et al. [48] described a patient who felt unwell 4 weeks following the first dose vaccination. The patient underwent kidney biopsy because serum creatinine was 2.7 mg/dl, and a positron emission tomography scan showed findings suggestive of vasculitis. Histopathology revealed diffuse interstitial edema with noncaseating granulomas around small vessels. Serum creatinine returned to normal within 4 weeks after methylprednisolone treatment.

Chavarot et al. reported [97] that a 66-year-old kidney transplant patient underwent a 1-year protocol graft biopsy 8 weeks after the second vaccine dose of COVID-19 vaccination, and kidney biopsy staining revealed the presence of MN. Notably, staining was retrospectively negative in the 3-month protocol graft biopsy and in the native kidney nephrectomy specimen. The diagnosis of de novo post-transplant MN was firmly established.

Fulchiero et al. [98] present a 21-year-old male with a history of FSGS who underwent kidney transplant at 13 years of age and was stable for the following 8 years. He presented with edema, and kidney biopsy showed FSGS recurrence following two doses of COVID-19 vaccination.

## 4. Discussion

Some cases of renal side effects were observed following COVID-19 vaccination. This narrative review shows that most patients received mRNA vaccines associated with postvaccination kidney disease development. This can be attributed to the fact that Pfizer and Moderna vaccines have been more accessible and widely used under emergency authorization by the FDA, probably due to cell-mediated and antibody-mediated immune responses generated by the mRNA vaccines.

The approved Pfizer and Moderna mRNA vaccines contain purified modified mRNA and lipid nanoparticle (LNP) delivery systems, while AstraZeneca and Johnson vaccines contain adenovirus (AdV) vector systems [99,100,101,102]. The design of a vaccine requires a pathogen-specific immunogen and an adjuvant to stimulate immunity. An optimal adjuvant provides the necessary second signal for T-cell activation without inducing systemic inflammation. Typically, following mRNA vaccination, the mRNA serves as both immunogen and adjuvant and is recognized by various endosomal and cytosolic innate sensors and taken up by dendritic cells. Cytosolic sensing, such as melanoma differentiation-associated protein 5 (MDA5) and retinoic acid-inducible gene I (RIG-I), binds to single-stranded RNA and double-stranded RNA, while Toll-like receptor (TLR) 3/7 binds to single-stranded RNA in the endosome, resulting in interferon regulatory factor 3/7 and nuclear factor κB activation and the production of type I interferon (IFN-I) and multiple inflammatory mediators [103,104]. Unlike mRNA vaccination, AdV vector vaccines also contain inherent adjuvant properties, which reside with the virus particle that encases the DNA encoding the immunogen. Following injection, the particles target dendritic cells and macrophages and bind to TLR9, resulting in IFN-I secretion [105]. IFN-I induces the differentiation of CD4+ and CD8+ effector T cells, which thereby stimulates the production of inflammatory and cytotoxic mediators and activated CD4+ T follicular helper cells stimulate B-cell differentiation into plasma cells that secrete antibodies (Figure 2). The activation of antigen-specific effector T cells peaks one week after vaccination [106]. From our review, more than half of the patients with MCD were observed following the first dose of the COVID-19 vaccination, and the median time for symptom onset was seven days, which supports the hypothesis for the generation of T-cell-mediated injury in MCD, and other studies demonstrated cellular immune responses approximately 1 week after viral infection [107].

Furthermore, compared with conventional vaccines, mRNA vaccines provoke a stronger reaction of CD4+ T and CD8+ T cells and higher production of cytokines [103,108]. The resulting permeability factors can directly affect podocytes and alter the glomerular permeability barrier, leading to MCD [106].

IgA nephropathy is the most common form of primary glomerulonephritis worldwide and is characterized by diffuse mesangial deposition of immunoglobulin A1 (IgA1) in glomeruli [109]. The pathogenesis of IgAN is still unclear, and several studies have reported that IgAN is a condition with several hypothetical pathological mechanisms, including genetic factors influencing the encoding galactosylation of IgA1 and environmental triggers, such as bacterial or viral infection and alteration of microbiota and food antigens [110]. In recent years, a multihit mechanism has been widely accepted. Specifically, increased synthesis of galactose-deficient IgA1 as the first hit, production of antiglycan IgG and/or IgA1 autoantibodies as the second hit, and poorly galactosylated IgA1 and antiglycan IgG autoantibodies form immune complexes and deposit in the mesangial area of the glomerulus from the subsequent hits [111] The deposition of IgA1 immune complexes causes mesangial cell, podocyte, and tubular epithelial cell damage, leading to end-stage renal disease in patients with IgAN.

Although a previously reported influenza vaccine was associated with IgAN in both native and transplanted kidneys, IgAN is not frequently reported following other vaccinations [112]. However, IgAN is the second most common renal side effect after COVID-19 vaccination, and the explanation has not been fully established. In this review, among 29 patients with new-onset IgAN, 12 patients showed a history of hematuria, which could suggest that IgAN is pre-existing and exacerbated by vaccination. In addition, almost 90% of patients developed gross hematuria and proteinuria after the second dose. One hypothesis is that the production of excess anti-glycan antibodies has a cross-reaction with pre-existing poorly galactosylated IgA1 following COVID-19 vaccination. Furthermore, similar to what was reported for influenza vaccines, the COVID-19 mRNA vaccine relates to a spike in IgA and IgG production in healthy adults and further increases after the second vaccination, while vaccination also stimulates robust T-helper cell and B-cell responses [113]. Therefore, the production of excess pathogenic IgA is another explanation. Generally, within 2 days, approximately 30% of patients experience systemic symptoms such as fever and pain after vaccination, which suggests systemic cytokine-mediated attack. Thus, another explanation is that the receptor binding domain of the SARS-CoV-2 spike protein may act as a superantigen, activate the immune system and cause cytokine storms in which inflammatory factors, such as IL-6 and IL-10, rise sharply [8].

It has been reported that ANCA-associated vasculitis occurs after injection of influenza and rabies vaccines based on viral mRNAs [114]. Moreover, a significantly reduced ANCA response after treatment with ribonuclease vaccines was observed. This encourages the question of whether ANCA vasculitis and RNA vaccines have a direct relationship. Although the occurrence of ANCA vasculitis disease after COVID-19 vaccination has been described, whether it more frequently occurs following mRNA vaccines is required to establish further real-world study. According to the report, a hyperactive immune and an autoimmune response toward SARS-CoV-2 cause ANCA and anti-GBM vasculitis [115,116,117]. Booster COVID-19 mRNA vaccination primes a notably enhanced innate immune response compared with primary immunization [118]. The heightened innate immune response following booster vaccination could be responsible for triggering the observed PR3 and MPO autoantibodies [51]. In ANCA-associated vasculitis, TLRs play crucial roles in initiating autoimmunity and inflammation, which are primarily attributed to TLR-2-induced Th17 autoimmunity, while TLR-9 promotes Th1 autoimmunity [119]. Interestingly, the activation of TLR2 in the immunodominant cytotoxic T lymphocyte response to the spike glycoprotein of SARS-CoV-2, which is also produced by COVID-19 vaccines, was described [120].

AIN is one of the leading causes of acute kidney injury, histopathologically characterized by the presence of inflammatory cell infiltrate in the interstitium, resulting in local edema and destruction of tubular basement membrane and interstitial architecture. Many causes of AIN are known, the most frequently triggered by immuno-allergic reactions to drug therapy that mainly involve nonsteroidal anti-inflammatory drugs and antibiotics; in addition, infections, toxins, and vasculitis can induce AIN [121]. In the review, all twelve patients who did not take any over-the-counter medications, including nonsteroidal anti-inflammatory drugs or herbal remedies, developed AIN after COVID-19 vaccination and responded well to steroid treatment. The mechanisms underlying the association of AIN with COVID-19 vaccination remain elusive. However, molecular mimicry could be applied to demonstrate the association between AIN and COVID-19 vaccination. Molecular mimicry refers to a similarity between certain pathogenic elements contained in the vaccine and human proteins with consecutive immune cross-reactivity [122]. Mira et al. [87] reported a case of AIN development after two doses of Pfizer vaccine that considered a drug-induced hypersensitivity reaction. The lymphocyte transformation test was positive for both vaccine solution and polyethylene glycol (PEG) excipient-indicated T cells, particularly for PEG-specific T cells involved in a type IV hypersensitivity reaction.

Although these may be a causative mechanism resulting from COVID-19 vaccination, granulomatous vasculitis, membranous nephropathy, and lupus nephritis will be required to further research the validity of these associations.

Transplant recipients have not been included in vaccine trials to date. Therefore, vaccine safety, efficacy, and durability profiles have not been measured in these patients. The immunosuppressed condition in these patients may cause a lower anti-SARS-CoV-2 antibody response, depending on the period since transplantation, the intensity of immunosuppression, and the type of transplantation. There would appear to be a slight chance of stimulating immunologic rejection reactions via the vaccination-induced immune response.

The first core issue is proof with certainty that the COVID-19 vaccine resulted in the development of new kidney disease or relapse of kidney disease. The time point of onset and exclusive diagnosis are useful for clinical diagnosis, and renal biopsy may be considered. Most new cases treated with steroid therapy can be improved. For patients with chronic kidney disease, closely observing the condition of basic diseases and monitoring disease activity after vaccination are important for identification of recurrence, and steroid combination immunosuppression can be rapidly relieved.

This review has certain limitations. First, not all patients experiencing renal side effect may have been included, because the search was performed in two databases and epidemiological investigations are lacking, so we cannot determine the true incidence of renal side effects after COVID-19 vaccination. Second, the mechanisms that we have discussed about the vaccine-related association only combine hypotheses from case reports, which have not been proven.

## 5. Conclusions

In conclusion, some studies reporting renal side effects appear to be associated with COVID-19 vaccination and we discussed the likely immune-mediated hypotheses of kidney disease following COVID-19 vaccination. Further research is warranted to better understand the causes and mechanisms of kidney disease after COVID-19 vaccination. All we can recommend at this point is that if some symptoms, such as hematuria, foamy urine, and edema, can be detected in an early phase, patients will benefit from a timely treatment.

Healthcare professionals should keep a watchful eye for these side effects and recognize them early and treat them efficiently.

## Figures and Tables

**Figure 1 vaccines-10-01783-f001:**
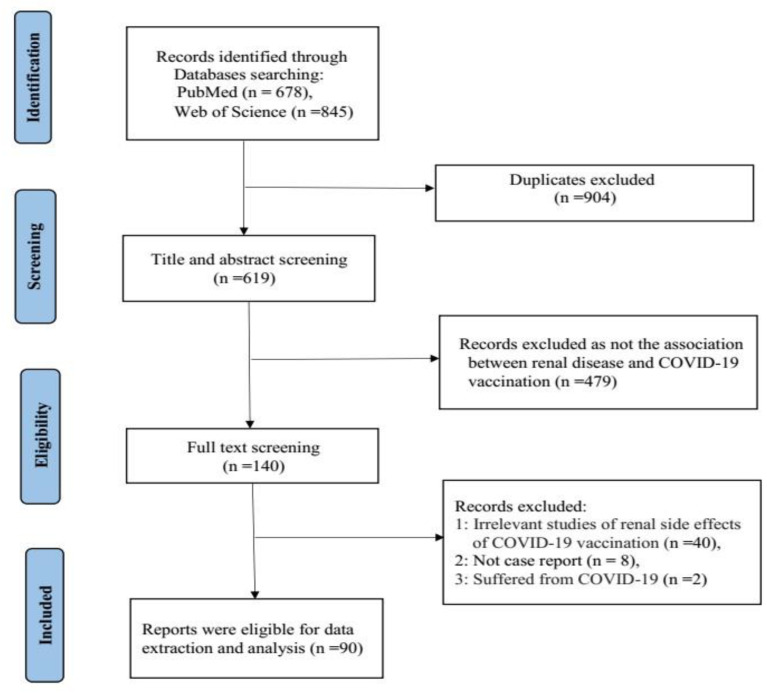
PRISMA flow diagram.

**Figure 2 vaccines-10-01783-f002:**
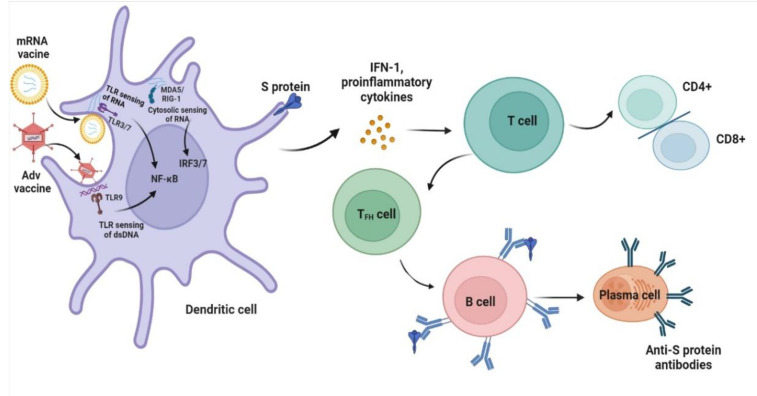
**Proposed mechanisms of minimal change disease caused by COVID-19 vaccination.** After mRNA vaccines and adenovirus (AdV) vector vaccines enter dendritic cells, high levels of S protein are produced. In addition, innate sensors such as endosomal T Toll-like receptor (TLR3 and TLR7) sensing of RNA and TLR9 binding double-stranded DNA (dsDNA) activate nuclear factor κB (NF-κB), and cytosolic sensing of melanoma differentiation-associated protein 5 (MDA5) and retinoic acid-inducible gene I (RIG-I) activate IFN regulatory factor 3/7 (IRF3/7), leading to the production of type I interferon (IFN-I) and proinflammatory cytokines. The resultant activated dendritic cells present antigens and costimulatory molecules to S protein-specific naive T cells and promote the adaptive immune response of T and B cells. The figure refers to the pathogenesis of a normal immune reaction in contact with pathogens by Sprent et al. [104].

**Table 1 vaccines-10-01783-t001:** Clinical characteristics of patients with MCD, IgAN, ANCA or AIN following COVID-19 vaccination.

Characteristics	MCD	IgAN	ANCA	AIN
	(*n* = 52)	(*n* = 48)	(*n* = 16)	(*n* = 12)
Age (years)	44 (14–83)	33 (12–79)	75 (29–82)	44 (12–77)
Male sex (%)	31 (60.0)	23 (47.9)	9 (56.3)	5 (41.7)
Medical history, *n* (%)				
None	15 (28.8)	6 (12.5)	5 (31.3)	6 (50.0)
Hypertension	5 (9.6)	3 (6.3)	3 (18.8)	1 (8.3)
Diabetes/dyslipidemia	6 (11.5)	2 (4.2)	1 (6.3)	3 (25.0)
Kidney disease/abnormal urine	10 (19.2)	31 (64.6)	0 (0.0)	0 (0.0)
Vaccine type, *n* (%)				
BNT-162b2 (Pfizer)	27 (51.9)	30 (62.5)	7 (43.8)	8 (66.7)
mRNA-1273 (Moderna)	14 (26.9)	15 (31.3)	6 (37.5)	3 (25.0)
Adenovirus vector (AstraZeneca)	7 (13.5)	2 (4.2)	3 (18.8)	1 (8.3)
Adenovirus vector (Janssen)	2 (3.8)	0 (0.0)	0 (0.0)	0 (0.0)
Inactivated vaccine (CoronaVac)	2 (3.8)	1 (2.1)	0 (0.0)	0 (0.0)
Cases, *n* (%)				
New cases	38 (73.1)	29 (60.4)	10 (62.5)	12 (100.0)
Relapsed cases	14 (26.9)	19 (39.6)	6 (37.5)	0 (0.0)
Symptoms occur after 1st or 2nd dose, *n* (%)				
1st dose	28 (53.8)	13 (27.1)	5 (31.3)	4 (33.3)
2nd dose	24 (46.2)	35 (72.9)	11 (68.8)	8 (66.7)
Onset, *n* (%)				
New cases 1st dose	22 (42.3)	10 (20.8)	2 (12.5)	4 (33.3)
New cases 2nd dose	16 (30.8)	19 (39.6)	8 (50.0)	8 (66.7)
Relapse cases 1st dose	6 (11.5)	3 (6.3)	3 (18.8)	0 (0.0)
Relapse cases 2nd dose	8 (15.4)	16 (33.3)	3 (18.8)	0 (0.0)
Symptoms, *n* (%)				
Gross hematuria	0 (0.0)	43 (89.6)	2 (12.5)	0 (0.0)
Proteinuria	7 (13.5)	7 (14.6)	2 (12.5)	0 (0.0)
Edema	40 (76.9)	0 (0.0)	1 (6.3)	0 (0.0)
Acute kidney injury/renal failure	3 (5.8)	10 (20.8)	3 (18.8)	12 (100.0)
Fever/headache/nausea/vomiting/ anorexia/diarrhea	5 (9.6)	14 (29.2)	4 (25.0)	5 (41.7)
Timing of symptom onset, *n* (%)				
1 day	4 (7.7)	18 (37.5)	3 (18.8)	0 (0.0)
2–7 days	25 (48.1)	20 (41.7)	4 (25.0)	4 (33.3)
> 7 days	23 (44.2)	9 (18.8)	9 (56.3)	8 (66.7)
Timing of symptom onset, days				
1st dose	7 (1–46)	4.0 (1–61)	7 (1–35)	14.5 (2–28)
2nd dose	8 (2–88)	2 (1–42)	14 (1–60)	14.0 (2–42)
Laboratory on presentation				
Baseline serum creatinine (mg/dl)	0.9 (0.7–1.2)	0.8 (0.5–1.3)	0.8 (0.7–2.6)	0.9 (0.9–1.0)
Serum creatinine (mg/dl)	1.2 (0.6–10.6)	1.2 (0.5–3.6)	1.8 (1.3–8.4)	4.5 (1.7–19.0)
Serum albumin (g/dl)	2.1 (0.5–4.7)	4.1 (1.9–5.3)	-	-
Urine protein (g/d)	14.0 (0.8–23.4)	1.5 (0.3–14.0)	-	-
UPCR (g/g)	10.7 (1.3–22.6)	1.3 (0.1–19.1)	-	0.9 (0.1–4.6)
Treatment, *n* (%)				
Immunosuppression (steroid)	39 (75.0)	12 (25.0)	2 (12.5)	7 (58.3)
Combination immunosuppression	5 (9.6)	4 (8.3)	6 (37.5)	0 (0.0)
Steroid + plasmapheresis/hemodialysis	3 (5.8)	3 (6.3)	8 (50.0)	4 (33.3)
Conservative management	2 (3.8)	24 (50.0)	0 (0.0)	1 (8.3)
Spontaneously	0 (0.0)	4 (8.3)	0 (0.0)	0 (0.0)
Not report	3 (5.8)	1 (2.1)	0 (0.0)	0 (0.0)
Outcome *, *n* (%)				
Response	41 (97.6)	37 (92.5)	10 (91.0)	12 (100.0)
Not response	1 (2.4)	3 (7.5)	1 (9.0)	0 (0.0)

Outcome * There were only 42 patients with MCD, 40 patients with IgAN and 11 patients with ANCA from the literature with follow-up. MCD, minimal change disease; IgAN, IgA nephropathy; ANCA, antineutrophil cytoplasmic autoantibody vasculitis; AIN, acute interstitial nephritis, UPCR, urine protein-to-creatinine ratio.

**Table 2 vaccines-10-01783-t002:** Summary of other cases of renal side effects following COVID-19 vaccination.

Authors	Age/Sex	Medical History	Vaccine	Timing of Symptom Onset	New/Relapse	Presenting Symptoms	Diagnosis	Baseline Scr (mg/dL)	Presentation Scr (mg/dL)	Presentation Urinalysis	Treatments	Outcomes
Da et al. [83]	70/M	None	Pfizer	Day 1 after 2nd dose	New	Edema	MN	NA	1.29	UTP: 4.4 g/d, RBC: 17/μL	Conservative	No spontaneous remission after 2 months
Fenoglio et al. [67]	82/F	NA	Pfizer	Day 88 after 2nd dose	New	Nephrotic syndrome	MN	NA	NA	NA	Glucocorticoids	NA
	67/F	NA	Pfizer	Day 89 after 2nd dose	New	Nephrotic syndrome	MN	NA	NA	NA	Rituximab	NA
	82/M	NA	Pfizer	Day 29 after 2nd dose	New	Nephrotic syndrome	MN	NA	NA	NA	Rituximab	NA
Klomjit et al. [41]	50/F	NA	Pfizer	4 weeks after 2nd dose	New	Joint pain and proteinuria	NELL-1 MN	0.84	0.7	UTP: 6.5 g/d, RBC: 3–10/ HPF	Conservative	Response; Scr was 0.7 mg/dL, RBC: <3/HPF, UTP was 0.4 g/d during last follow-up
	39/M	MN	Pfizer	1 week after 2nd dose	Relapse	Edema	PLA2R MN	0.91	1.13	UTP: 8.7 g/d, RBC: 3–10/ HPF	Tacrolimus	Response; Scr was 1.1 mg/dL, RBC:3–10/HPF, UTP was 5.7 g/d during last follow-up
	70/M	MN	Moderna	4 weeks after 2nd dose	Relapse	Edema	PLA2R MN	1.7	2.1	UTP: 16.6 g/d, RBC: <3/ HPF	Obinutuzumab	NA
Aydın et al. [55]	66/F	Primary MN, diabetes mellitus and hyperlipidemia	Sinovac	2 weeks after 1st dose	Relapse	Edema	PLA2R MN	NA	2.78	UPCR: 9.42 g/g	NA	NA
Dormann et al. [37]	20/F	None	Pfizer	Day 5 after 1st dose	New	Edema	FSGS	NA	0.47	UPCR:10.3 g/g	Prednisolone, diuretic and lipid-lowering	Partial remission with persisting proteinuria and hyperlipoproteinemia
Fenoglio et al. [67]	24/F	NA	Pfizer	Day 88 after 2nd dose	New	Nephrotic syndrome	FSGS (tip variant)	NA	NA	NA	Glucocorticoids	NA
Klomjit et al. [41]	29/F	FSGS	Pfizer	3 weeks after 2nd dose	Relapse	Edema	FSGS (tip variant)	0.6	0.6	UTP: 10 g/d, RBC: <3/ HPF	Steroid, tacrolimus	Response; Scr was 0.7 mg/dL, RBC: <3/HPF, UTP was 3.7 g/d during last follow-up
Gillion et al. [48]	77/M	None	AstraZeneca	4 weeks after 1st dose	New	Fever, night sweating, and anorexia	Granulomatous vasculitis	1.2	2.7	Normal proteinuria, no hematuria	Methylprednisolone	Scr normalized within 4 weeks
Sacker et al. [54]	Older/F	None	Moderna	2 weeks after 2nd dose	New	Fever, anorexia, nausea, and gross hematuria	Anti-GBM with mesangial IgA deposits	NA	7.8	UPCR: 1.9 g/g	Methylprednisolone, cyclophosphamid, plasmapheresis, and hemodialysis	No response; dialysis-dependent
Tan et al. [70]	60/F	Hyperlipidemia	Pfizer	Day 1 after 2nd dose	New	Gross hematuria	Anti-GBM nephritis	NA	6.11	UPCR: 7.58 g/g	Methylprednisolone, prednisolone, cyclophosphamide, plasma exchange	NA
Klomjit et al. [41]	77/M	NA	Pfizer	1 week after 1st dose	New	Hypertension	Atypical anti-GBM nephritis	1	1.8	UTP: 1.6 g/d, RBC: 51–100/ HPF	Prednisone, mycophenolate	No response; Scr was 2.9 mg/dL, RBC: 51–100/HPF, UTP was 0.3 g/d during last follow-up
Tuschen et al. [81]	42/F	LN	Pfizer	1 week after 1st dose	Relapse	Nephrotic syndrome	Class V LN	NA	NA	UTP: 8.4 g/d	Mycophenolate mofetil and prednisolone	Proteinuria improved
Kim et al. [91]	60/F	Oral corticosteroids for a skin rash	AstraZeneca	Several weeks after 2nd dose	New	Edema, proteinuria	Class III LN	0.74	1.81	UPCR: 4.82 g/g	Methylprednisolone, cyclophosphamide, prednisolone and hydroxychloroquine	Response; Scr was 0.93 mg/dL, UPCR: 1.64 g/g after ten days treatment
Zavala et al. [82]	22/F	None	AstraZeneca	1 week after 1st dose	New	Edema, proteinuria	Class V LN	NA	0.8	UTP: 12.6 g/d, UPCR: 11.0 g/g	Mycophenolate mofetil, glucocorticoids, hydroxychloroquine, and diuretics	Edema improved after 3 weeks follow-up

Abbreviations: M, male; F, female; HPF, high-powered field; MN, membranous nephropathy; FSGS, focal segmental glomerulosclerosis; GBM, glomerular basement membrane; SCr, serum creatinine; PLA2R, phospholipase A2 receptor; LN, lupus nephritis; NA, nonapplicable. UTP, 24-h urine protein; UPCR, urine protein-to-creatinine ratio.

## Data Availability

The data that support the findings of this study are available from the corresponding author upon reasonable request.

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
