# Peer review of "Renal Side Effects of COVID-19 Vaccination"

_vaccines, 2022, doi:10.3390/vaccines10111783_

Round 1

Reviewer 1 Report

  • The English language of the manuscript needs to be improved. In addition to grammatical mistakes, there are sections of the paper which are not fully coherent, and seem more like just a list of what is already presented in the Tables rather than a comprehensive presentation and overview of the state in the field.
  • Lines 130-131: This is not clear - why were only cases in adolescents and children searched for, if the results section states that the age range was 12-83 years. The title does not indicate this either.
  • Lines 130-131: How about databases where adverse events get reported? If not, this is a limitation of what is named a systematic review here. In addition, why only one database? This poses a risk of missing out on some possibly relevant articles.
  • Lines 132-133: This needs to be more specific - cases of what. What was newly diagnosed, what relapsed - be precise.
  • Lines 132-138: Was it possible to stratify the results you present here by vaccine the persons received?
  • Lines 163-174: This section is very vague and lacks description of Methods. Why was only Pubmed searched? What were the including criteria? What kind of studies did you consider? What were the excluding criteria?
  • Lines 167-168: What does it mean that you excluded repetitive articles? Duplicates can be excluded, but what do you mean by repetitive? Also, how were full-text articles excluded? This inputs a serious bias in this study which can not be justified - this must be corrected. How many of the articles did you exclude because you could not find full text?
  • Lines 194-196: Did all patients receive the same type of vaccine for their first dose and second dose?
  • Table 1: Medical history - None has "healthy" in the brackets - but, was there no record of medical history so it is none or none because the history was not remarkable for any disease or condition?
  • Table 2: What does it mean "summary of a few published cases" - how were these few selected, based on what criteria?
  • Line 116: Align the statements - some parts of the paper state that the number is not remarkable compared to the number of vaccinations, and here it is stated that the cases are considerable.
  • A figure depicting the literature search - flow diagram, should be added, with details of the number of articles found, included, excluded and detailed reasons.
  • Discussion: Discuss strengths and limitations of this systematic review.
  • Conclusions: Keep the conclusion to what you found in the review and what implications your findings have for future research, rather than discussing causal associations which a narrative review cannot. 

Author Response

  • The English language of the manuscript needs to be improved. In addition to
    grammatical mistakes, there are sections of the paper which are not fully coherent,
    and seem more like just a list of what is already presented in the Tables rather than a
    comprehensive presentation and overview of the state in the field

Answer: Thank the reviewer very much. We have amended it.

  • Lines 130-131: This is not clear - why were only cases in adolescents and children searched for, if the results section states that the age range was 12-83 years. The title does not indicate this either.

Answer: Thank the reviewer very much for pointing out written mistakes. We searched articles reporting renal adverse reactions including adults and children. We have amended it.

  • Lines 130-131: How about databases where adverse events get reported? If not, this is a limitation of what is named a systematic review here. In addition, why only one database? This poses a risk of missing out on some possibly relevant articles.

Answer: Thank the reviewer very much. A systematic literature search was conducted in the databases PubMed in our previous manuscripts. Now, we were authorized by Zhejiang University to visit the web of science. We searched in the databases ‘PubMed’ and ‘Web of Science’. Finally, 90 papers 130 cases developed renal side effects after COVID-19 vaccination were included in this systematic review and we reanalyzed the data (seee Table 1).

  • Lines 132-133: This needs to be more specific - cases of what. What was newly diagnosed, what relapsed - be precise.

Answer: Thank the reviewer very much. We have amended it as: A total of 130 cases reported a renal adverse reaction following COVID-19 vaccination from 90 articles were included in this review, of which 90 (69%) were new-onset kidney diseases while 40 (31%) were relapsed kidney diseases.

  • Lines 132-138: Was it possible to stratify the results you present here by vaccine the persons received?

Answer: Thank the reviewer very much. We have amended it as: The most frequent renal side effects of COVID-19 vaccination were minimal change disease (52 cases), IgA nephropathy (48 cases), antineutrophil cytoplasmic autoantibody vasculitis (16 cases) and acute interstitial nephritis (12 cases). Other renal side effects occur at a much lower frequency.

  • Lines 163-174: This section is very vague and lacks description of Methods. Why was only Pubmed searched? What were the including criteria? What kind of studies did you consider? What were the excluding criteria?

Answer: Thank the reviewer very much. We add the Prisma flow diagram (see Fig 1).The study selection process was carried out using the Preferred Reporting Items for Systematic Reviews and Meta Analyses (PRISMA) guideline. We were authorized by Zhejiang University to visit the web of science, and we used search engines of PubMed’ and Web of Science. After the search was complete, and all duplicates were removed, then the abstracts of the remaining articles were reviewed to ensure they address the review question.

Title and abstract screening: Studies are included if they satisfy all questions as follows: Does the study focus on SARS-CoV-2? Does the study present COVID-19 vaccines? Does the study present adverse reactions to COVID-19 vaccines? Does the study present the renal side effects of COVID-19 vaccines?

Full-text screening: Studies are included if they satisfy all questions as follows: What is the type of renal side effects after COVID-19 vaccines (new onset or relapsed)? What is the type of study (journal article, book chapter, report, review, letter)? Whether a renal biopsy was performed? What is the kidney histopathology?

  • Lines 167-168: What does it mean that you excluded repetitive articles? Duplicates can be excluded, but what do you mean by repetitive? Also, how were full-text articles excluded? This inputs a serious bias in this study which can not be justified - this must be corrected. How many of the articles did you exclude because you could not find full text?

Answer: Thank the reviewer very much. In our previous manuscripts, we use search engines of databases PubMed. There are several articles published as case report, however, this case is also included in some series of case reports. Thus, we excluded these repetitive articles. Full text exclusion is because we have no authorization to obtain the full text and several articles have only abstracts but no full text were also excluded. We add the Prisma flow diagram (see Fig 1). Now, we add the Prisma flow diagram (see Fig 1).

  • Lines 194-196: Did all patients receive the same type of vaccine for their first dose and second dose?

Answer: Thank the reviewer very much. A total of 78 patients developed renal side effects after receiving the second dose of COVID-19 vaccines. However, almost articles did not describe whether the second dose of COVID-19 vaccines is the same type of the first dose.

  • Table 1: Medical history - None has "healthy" in the brackets - but, was there no record of medical history so it is none or none because the history was not remarkable for any disease or condition?

Answer: Thank the reviewer very much for pointing out our mistakes. Strictly speaking, none does not represent healthy. We have deleted the “healthy” in Table 1.

  • Table 2: What does it mean "summary of a few published cases" - how were these few selected, based on what criteria?

Answer: Thank the reviewer very much. This sentence described is not precise.

Table 2 shows other renal side effects of membranous nephropathy, focal segmental glomerulosclerosis; anti-glomerular basement membrane, lupus nephritis following COVID-19 vaccination. We have amended it.

  • Line 116: Align the statements - some parts of the paper state that the number is not remarkable compared to the number of vaccinations, and here it is stated that the cases are considerable.

Answer: Thank the reviewer very much for pointing out our mistakes. We have amended it.

  • A figure depicting the literature search - flow diagram, should be added, with details of the number of articles found, included, excluded and detailed reasons.

Answer: Thank the reviewer very much. We add the Prisma flow diagram (see Fig 1).

  • Discussion: Discuss strengths and limitations of this systematic review.

Answer: Thank the reviewer very much. We add the discussion and limitations.

  • Conclusions: Keep the conclusion to what you found in the review and what implications your findings have for future research, rather than discussing causal associations which a narrative review cannot. 

Answer: Thank the reviewer very much. We have amended it.

Reviewer 2 Report

Thank you for invitation to review this manuscript. The authors have tried to bind the results on renal side effects of COVID-19 vaccines from previously published studies. They claimed to have systematic review methods, but these methods were not fairly described in the manuscript. The methodology of this study is not according to the standards of PRISMA and other guidelines on systematic review. The authors must provide information on search strategies, search terms, databases used, results of each search, study selection criteria, risk of bias in studies, methods of extracting data, etc. I am not sure that the protocol of this review was previously designed and approved by the authors.

It is unclear why the authors focused on only four adverse events, or these are the only adverse events on renal problems to date. If so, the authors need to clarify that the literature search only revealed four side effects, and there are no other renal side effects linked with the use of vaccines.

The type of studies included in this review is not clear. Does it seem that all the results and conclusions are based on the information extracted from the case reports? As I know, there are also some cross-sectional or cohort studies in this regard. It needs clarification.

Please add the Prisma flow diagram

please add the PRISMA check list as a supplementary file.

Since this manuscript does not follow rigorous methodology, the results and conclusions are accompanied by several potential biases.

Author Response

Thank you for invitation to review this manuscript. The authors have tried to bind the results on renal side effects of COVID-19 vaccines from previously published studies. They claimed to have systematic review methods, but these methods were not fairly described in the manuscript. The methodology of this study is not according to the standards of PRISMA and other guidelines on systematic review. The authors must provide information on search strategies, search terms, databases used, results of each search, study selection criteria, risk of bias in studies, methods of extracting data, etc. I am not sure that the protocol of this review was previously designed and approved by the authors.

Answer: Thank the reviewer very much. We add the Prisma flow diagram (see Fig 1).The study selection process was carried out using the Preferred Reporting Items for Systematic Reviews and Meta Analyses (PRISMA) guideline. We were authorized by Zhejiang University to visit the web of science, and we used search engines of PubMed’ and Web of Science. After the search was complete, and all duplicates were removed, then the abstracts of the remaining articles were reviewed to ensure they address the review question.

Title and abstract screening: Studies are included if they satisfy all questions as follows: Does the study focus on SARS-CoV-2? Does the study present COVID-19 vaccines? Does the study present adverse reactions to COVID-19 vaccines? Does the study present the renal side effects of COVID-19 vaccines?

Full-text screening: Studies are included if they satisfy all questions as follows: What is the type of renal side effects after COVID-19 vaccines (new onset or relapsed)? What is the type of study (journal article, book chapter, report, review, letter)? Whether a renal biopsy was performed? What is the kidney histopathology?

It is unclear why the authors focused on only four adverse events, or these are the only adverse events on renal problems to date. If so, the authors need to clarify that the literature search only revealed four side effects, and there are no other renal side effects linked with the use of vaccines.

Answer: Thank the reviewer very much. Table 1 shows the most frequent renal side effects of COVID-19 vaccination were minimal change disease (52 cases), IgA nephropathy (48 cases), antineutrophil cytoplasmic autoantibody vasculitis (16 cases) and acute interstitial nephritis (12 cases). Table 2 shows other renal side effects of membranous nephropathy, focal segmental glomerulosclerosis; anti-glomerular basement membrane, lupus nephritis following COVID-19 vaccination occur at a much lower frequency. And, we present two cases of kidney transplant recipient developed renal side effects following COVID-19 vaccination.

The type of studies included in this review is not clear. Does it seem that all the results and conclusions are based on the information extracted from the case reports? As I know, there are also some cross-sectional or cohort studies in this regard. It needs clarification.

Answer: Thank the reviewer very much. We searched case and case series reports that were original articles which convincingly reported a renal adverse reaction. We then extracted patient demographics (age and sex), medical history, vaccine type, number of vaccine doses given, baseline characteristics, laboratories upon presentation, onset of symptoms, timing of symptom onset, treatments, and outcomes. We have amended it in methods section line 164 according to your advice.

Please add the Prisma flow diagram

Answer: Thank the reviewer very much. We add the Prisma flow diagram (see Fig 1)..

please add the PRISMA check list as a supplementary file.

Answer: Thank the reviewer very much. We add the PRISMA check list as a supplementary file.

Since this manuscript does not follow rigorous methodology, the results and conclusions are accompanied by several potential biases

Answer: Thank the reviewer very much. We add the Prisma flow diagram (see Fig 1).The study selection process was carried out using the Preferred Reporting Items for Systematic Reviews and Meta Analyses (PRISMA) guideline. Due to the limitation of database access authorization, we only use PubMed and Web of Science database. Although most articles retrieved from the two databases are consistent, the results and conclusions are accompanied by several potential biases. We add it in limitations of this systematic review.

Reviewer 3 Report

Dear authors, thank you very much for your work.  It summarises the renal adverse events which could be associated with vaccination against COVID19. I have the following comments which, in my opinion, could strengthen your work. 

1. Articles and literature on adverse renal events in patients with transplanted kidneys should be added. A comparison among native and transplanted kidneys is therefore of importance to compare the differences in renal injury patterns.

2. The simple demonstration of facts is inadequate. I think it could be useful for the reader to understand the importance of each point you make. A commentary summarizing and outlining the most important ideas within each paragraph and not only in the conclusions section would be very helpful.

3. Table 2 is complex to understand. Please simplify it. Since Proteinuria, Outcome, Treatment, Diagnosis, Urinalysis, Presenting symptoms, Onset, and serum creatinine are reported in Table 1, one could omit those in table 2.

All of my best regards.

Author Response

  1. Articles and literature on adverse renal events in patients with transplanted kidneys should be added. A comparison among native and transplanted kidneys is therefore of importance to compare the differences in renal injury patterns.

Answer: Thank the reviewer very much. In our previous manuscripts, transplanted kidneys were exclude. According to your advice, we add it in results section line 313. We present two cases of kidney transplant recipient developed renal side effects following COVID-19 vaccination.

  1. The simple demonstration of facts is inadequate. I think it could be useful for the reader to understand the importance of each point you make. A commentary summarizing and outlining the most important ideas within each paragraph and not only in the conclusions section would be very helpful.

Answer: Thank the reviewer very much. We have amended it.

  1. Table 2 is complex to understand. Please simplify it. Since Proteinuria, Outcome, Treatment, Diagnosis, Urinalysis, Presenting symptoms, Onset, and serum creatinine are reported in Table 1, one could omit those in table 2.

Answer: Thank the reviewer very much. We extracted patient demographics (age and sex), medical history, vaccine type, number of vaccine doses given, baseline characteristics, laboratories upon presentation, onset of symptoms, timing of symptom onset, treatments, and outcomes. Table 1 shows the most frequent renal side effects of COVID-19 vaccination were minimal change disease (52 cases), IgA nephropathy (48 cases), antineutrophil cytoplasmic autoantibody vasculitis (16 cases) and acute interstitial nephritis (12 cases). Table 2 shows other renal side effects of membranous nephropathy, focal segmental glomerulosclerosis; anti-glomerular basement membrane, lupus nephritis following COVID-19 vaccination occur at a much lower frequency.

Reviewer 4 Report

n their narrative review, Zang et al summarized the existing evidence on kidney side effects of SARS-CoV-2 vaccination.
The review is well written and exhaustive, but I have some comments
- in the introduction page 1 lines 145-146: there actually are therapeutic agents for COVID 19 (nirmatrelvir/ritonavir, remdesivir, molnupiravir)
- in the methods section:
It would be interesting to know if these patients had suffered from COVID-19 and if yes to investigate whether there are differences between the two group of patients

Author Response

- in the introduction page 1 lines 145-146: there actually are therapeutic agents for COVID 19 (nirmatrelvir/ritonavir, remdesivir, molnupiravir)

Answer: Thank the reviewer very much for pointing out our mistakes. We have amended it as: With the ongoing COVID-19 pandemic, vaccination programmes are being rolled out worldwide to prevent COVID-19 and alleviate the severity of the disease.

- in the methods section:
It would be interesting to know if these patients had suffered from COVID-19 and if yes to investigate whether there are differences between the two group of patients

Answer: Thank the reviewer very much. The articles that were repetitive or not full text were excluded and patients had suffered from COVID-19 were also excluded. This systematic review provides insight into native and transplanted kidneys developed renal side effects following COVID-19 vaccination and discuss the plausible mechanism of action triggered by COVID-19 vaccination.

Reviewer 5 Report

“Renal side effects of COVID-19 vaccination”. This article includes the case study of patients who has undergone side effects after vaccination. This article has went through the deep study and covered the facts of side effects which the world has ignored due to the pandemic situation. The emergency authorization lead to save the people. But more case studies should be done and understand the mechanism of these adverse effects in long run. Though the case study was good initiative there are still some shortcomings.

1.      In the case study can Author put their previous medical history and present side effects

2.      To prove that these vaccines has some adverse effects in different stream, which vaccines has the more on which type of individuals (Normal/Pre medical)

3.      It would be highly appreciated if the author could take the patients who are pregnant and does it have any side effects on the babies (Renal) or Moms

4.      Overall the case study less than 30 years old gives the optimal data. Since the age after 35 will have more medical issues. We can just conclude the mechanism to the side effects.

Author Response

  1. In the case study can Author put their previous medical history and present side effects

Answer: Thank the reviewer very much. We extracted patient demographics (age and sex), medical history, vaccine type, number of vaccine doses given, baseline characteristics, laboratories upon presentation, onset of symptoms, timing of symptom onset, treatments, and outcomes. Table 1 shows the most frequent renal side effects of COVID-19 vaccination were minimal change disease (52 cases), IgA nephropathy (48 cases), antineutrophil cytoplasmic autoantibody vasculitis (16 cases) and acute interstitial nephritis (12 cases). Table 2 shows other renal side effects of membranous nephropathy, focal segmental glomerulosclerosis; anti-glomerular basement membrane, lupus nephritis following COVID-19 vaccination occur at a much lower frequency.

  1. To prove that these vaccines has some adverse effects in different stream, which vaccines has the more on which type of individuals (Normal/Pre medical)

Answer: Thank the reviewer very much. Renal side effects develop after any of the commercially available COVID-19 vaccinations, but 56% (72/128) of patients received the BNT162b2 (Pfizer) vaccine, followed by 30% (38/128) receiving the mRNA-1273 (Moderna) vaccine. In addition, 10.2% (13/128) received adenovirus vector (AstraZeneca) vaccine, 1.5% (2/128) received adenovirus vector (Janssen) vaccine, and another 2.3% (3/128) received inactivated vaccine (CoronaVac). Of these, 39% (50/128) of patients developed symptoms after the first dose, while 61% (78/128) of patients developed symptoms after the second dose. For MCD, approximately 28.8% of patients did not have any medical history of chronic illness, while 21% of patients had a history of hypertension, diabetes, or dyslipidemia. For IgAN, approximately 64.6% of cases had a history of abnormal urine or kidney disease. Twelve patients with a median age of 44 (12-77) years developed new-onset AIN, all twelve patients presented with acute kidney injury.

  1. It would be highly appreciated if the author could take the patients who are pregnant and does it have any side effects on the babies (Renal) or Moms

  Answer: Thank the reviewer very much. It would be interesting to know if pregnant and does it have any side effects on the babies (Renal) or Mom, and it could strengthen our work. However, we did not find the study about renal side effect of pregnant in our search strategy and study selection. Little is known about the efficacy and safety profile of SARS‐CoV‐2 vaccines in pregnant. Different countries have different policies for vaccinating pregnant women against Covid‐19. Vaccinated pregnant mothers can pass the IgG antibodies produced to their offspring, with one case study reporting vertical transmission. It was also shown that the transplacental transfer of vaccine‐induced antibodies to the newborn is more likely if the mother is vaccinated in the third trimester. A study reported the safety and immunogenicity of the Pfizer/BioNTech vaccine in pregnant women. It shown that there were no additional adverse effects of vaccination in pregnant compared with non-pregnant women

  1. Overall the case study less than 30 years old gives the optimal data. Since the age after 35 will have more medical issues. We can just conclude the mechanism to the side effects

 Answer: Thank the reviewer very much. The search of articles reporting renal adverse reactions including adults and children were conducted. A total of 130 cases reported a renal adverse reaction following COVID-19 vaccination from 90 articles were included in this review. This narrative review was conducted to collect the renal side effects in the published data and discuss the plausible mechanism of action triggered by COVID-19 vaccination.

Round 2

Reviewer 1 Report

Thanks to the authors for revising their paper. However, there are issues that remain to be addressed: - Add the citation for PRISMA guidelines in the methodology section. - The added PRISMA flow diagram is not complete - authors did not list any reason for exclusion, and the diagram explicitly requires to state how many papers were excluded and why.
- Instead of saying repetitive articles, state that duplicates were removed and that papers reporting the same case were excluded too.
- The information given in the response letter regarding the inclusion and exclusion criteria needs to be added to the paper. And it is not enough to only say what the question was for screening but what was the answer for inclusion and exclusion. For example, question you stated is: What is the type of study (journal article, book chapter, report, review, letter)? But, which of these did you include? A clear section describing inclusion and exclusion criteria must be added to the paper.  - Did the paper now include papers which were before not available in full-text? It is not clear. Authors added that they searched Web of Science, does this mean that they now added articles? I see they increased some numbers in the revised manuscript. - The authors wrote in the response that many papers did not state whether the second dose of the vaccine (after which a renal side effect was recorded) was the same type as the first dose - but did not add this information to the paper too. - Lines 230-234: References need to be added for this text.  - Please once again check the English language of the manuscript, particularly in the newly added text, as there are still mistakes in the text of the paper.

Reviewer 2 Report

Thank you for clarification.

Author Response

We thank the reviewer very much. We have rechecked the language and form of the manuscript.

Reviewer 3 Report

Dear authors, thank you very much for this improved version of your work.

Please check the manuscript for minor spelling/syntax errors.

All of my best regards.

Author Response

(The authors gave the same response as above.)

Round 3

Reviewer 1 Report

Thank you for revising the paper. The following was not addressed:   - You did not add all information you had in your previous response letter when you explained inclusion/exclusion criteria - the article types, the renal biopsy/histopathology - what was included or excluded? Add this as a sentence to Methodology. - Conclusion: Needs to be rephrased. First sentence can be deleted or modified. Start the conclusion by saying what you found. Instead of saying primary disease say kidney disease in order to be clear.

Author Response

Thank you for revising the paper. The following was not addressed:   - You did not add all information you had in your previous response letter when you explained inclusion/exclusion criteria - the article types, the renal biopsy/histopathology - what was included or excluded? Add this as a sentence to Methodology.

Answer: We thank the reviewer very much. We have amended it as:

We only selected case or case series reports, other types of articles were excluded

Case or case series reporting new onset or relapse kidney histopathology in both native and transplanted kidneys following COVID-19 vaccination were included, and patients who had suffered from COVID-19 were excluded.

 - Conclusion: Needs to be rephrased. First sentence can be deleted or modified. Start the conclusion by saying what you found. Instead of saying primary disease say kidney disease in order to be clear

Answer: We thank the reviewer very much. We have amended it as:

In conclusion, some studies reporting renal side effects appears to be associated with COVID-19 vaccination and we discussed the likely immune-mediated hypotheses of kidney disease following COVID-19 vaccination. Further research is warranted to better understand the causes and mechanisms of kidney disease after COVID-19 vaccination. All we can recommend at this point is that if some symptoms, such as hematuria, foamy urine and edema, can be detected in an early phase, patients will benefit from a timely treatment. Healthcare professionals should keep a watchful eye for these side effects and recognize them early and treat them efficiently.
